# The Alleviation of Dextran Sulfate Sodium (DSS)-Induced Colitis Correlate with the log*P* Values of Food-Derived Electrophilic Compounds

**DOI:** 10.3390/antiox11122406

**Published:** 2022-12-05

**Authors:** Xiang-Rong Cheng, Bu-Tao Yu, Jie Song, Jia-Hui Ma, Yu-Yao Chen, Chen-Xi Zhang, Piao-Han Tu, Mitchell N. Muskat, Ze-Gang Zhu

**Affiliations:** 1School of Food Science and Technology, Jiangnan University, Wuxi 214122, China; 2National Engineering Research Center for Functional Food, Jiangnan University, Wuxi 214122, China; 3School of Pharmacy, University of California San Francisco, San Francisco, CA 94143, USA; 4Jinhua Academy of Agricultural Sciences, Jinhua 321000, China

**Keywords:** electrophilic compound, colitis, molecular characteristic, log*P*

## Abstract

Food-derived electrophilic compounds (FECs) are small molecules with electrophilic groups with potential cytoprotective effects. This study investigated the differential effects of six prevalent FECs on colitis in dextran sodium sulfate (DSS)-induced mice and the underlying relationship with molecular characteristics. Fumaric acid (FMA), isoliquiritigenin (ISO), cinnamaldehyde (CA), ferulic acid (FA), sulforaphane (SFN), and chlorogenic acid (CGA) exhibited varying improvements in colitis on clinical signs, colonic histopathology, inflammatory and oxidative indicators, and Nrf2 pathway in a sequence of SFN, ISO > FA, CA > FMA, CGA. Representative molecular characteristics of the “penetration-affinity–covalent binding” procedure, log*P* value, Keap1 affinity energy, and electrophilic index of FECs were theoretically calculated, among which log*P* value revealed a strong correlation with colitis improvements, which was related to the expression of Nrf2 and its downstream proteins. Above all, SFN and ISO possessed high log*P* values and effectively improving DSS-induced colitis by activating the Keap1–Nrf2 pathway to alleviate oxidative stress and inflammatory responses.

## 1. Introduction

Inflammatory bowel disease is a chronic idiopathic bowel disease that includes ulcerative colitis (UC) and Crohn’s disease (CD) [1]. At the beginning of the 21st century, inflammatory bowel disease became a global disease, with more than 17 million people in North America and Europe suffering from the disease, and its incidence is increasing in Asia, Africa, and South America [2]. The environment, immune and genetic factors are the main causes of UC, but its pathogenesis is still unclear [3]. UC mainly affects the mucosa and submucosa of the colon, resulting in inflammation and eventually damage to the entire colon, most commonly manifesting as blood in the stool and diarrhea [4]. Not only the inflammatory response but also oxidative stress occurs in the pathogenesis of UC, and often a cycle of inflammation and oxidative stress occurs, leading to a deepening of the severity of UC [5]. Oxidative stress occurs primarily in the inflammatory response, mainly because inflammatory cells, neutrophils, and macrophages produce large amounts of free radicals. Their excessive accumulation in the body leads to oxidative damage to intracellular macromolecules, such as proteins and DNA, and damages cell membranes [6]. In addition, the excess production of free radicals can further activate the Nrf2 signaling pathway [7]. It is well known that the Nrf2 pathway is a classical signaling pathway against oxidative stress [8]. Under basal conditions, Keap1 binds to Nrf2 in the cytoplasm and promotes ubiquitous degradation of Nrf2, resulting in the maintenance of low intracellular Nrf2 levels. When cells are subjected to oxidative stress, Nrf2 is released into the nucleus. It upregulates the transcript levels of antioxidant response element (ARE)-related genes, such as oxygenase-1 (HO-1), reduced coenzyme (NAD(P)H), and reduced coenzyme II (NQO1) [9]. It has been shown that Nrf2 activators protect against dextran sodium sulfate (DSS)-induced colitis in mice by lowing free radicals levels [10]. In contrast, Nrf2-deficient mice induced by DSS were found to be more susceptible to colitis, as evidenced by a significant reduction in colon length, severe histological changes, inflammatory cell infiltration, and elevated levels of pro-inflammatory cytokines [11]. Therefore, targeted inhibition of the inflammatory response and activation of the Keap1–Nrf2 pathway are considered as effective ways to regulate the redox homeostasis and intervene in UC.

Electrophilic compounds are a group of small molecules characterized by electron-deficient groups and prone to covalently bind with electron-rich biomolecules (such as proteins and DNA) [12]. Food-derived electrophilic compounds (FECs) are one of the categories which are abundant and frequently contained in daily food. Electrophilic compounds with various structures could reversibly or irreversibly bind to nucleophilic sites of cellular macromolecules, especially the sulfhydryl sites of functional proteins, mainly leading to the following two kinds of different effects: (i) detoxification, antioxidant, and anti-inflammatory cytoprotective effects; FECs belonging to this category, such as sulforaphane (SFN) and isoliquiritigenin (ISO), could promote the stability and nuclear translocation of Nrf2 and mediate the expression of phase II detoxification enzymes HO-1 and NQO1 [13,14]; (ii) biotoxicity; for example, acrolein can deplete glutathione (GSH) and disrupt the mitochondria of the cell, thus mediating cellular oxidative stress and leading to a variety of defects that may cause diseases [15]. Although the chemical structures of electrophilic compounds are various, one common property is the chemical reactivity with sulfhydryl groups, targeting modification of reactive cysteine thiols on Keap1, acting as an inducer sensor [16]. SFN, an isothiocyanate prevalent in cruciferous vegetables, could activate the Keap1–Nrf2 pathway to exert antioxidant and anti-inflammatory effects by inhibiting the expression of genes encoding major pro-inflammatory cytokines or upregulating leukotriene B4 dehydrogenase encoding Nrf2 target genes [17]. ISO is a chalcone found in licorice, which has anti-inflammatory, anti-atherosclerotic and anti-cancer effects, as well as activating the Keap1–Nrf2 pathway to reduce the damage caused by intestinal inflammation [18].

Molecular characteristics, also known as molecular descriptors, are symbolic or numerical indicators that translate the physicochemical properties contained in the molecular structure of a compound by logical or mathematical methods [19]. Currently, many different kinds of molecular descriptors have been developed, such as carbon atom number, lipid–water partition coefficient, molecular weight, properties calculated from 2D and 3D structures, and properties based on quantum mechanics, etc. [20]. In quantitative structure–activity relationship studies, molecular descriptors are often used to predict properties such as bioactivity resulting from the chemical structure of compounds [21].

A previous study on the reactivity of electrophilic compounds with thiols suggested a noncovalent affinity followed by a covalent bond formation process [22]. The ligand interacts reversibly with the specific pocket of the target protein to form an intermediate. The electrophilic part of the ligand in the intermediate complex will covalently interact with the nucleophilic amino acid residues of the target protein through addition, substitution, oxidation, and other reactions to form a covalent complex. However, there are limitations in describing the reaction between electrophilic compounds and their protein targets in terms of affinity and electrophilic capability. These molecular characteristics do not represent the process by which electrophilic compounds enter the organism and reach the “destination” of the reaction. Therefore, we propose that the electrophilic compounds penetrates through the cells first, creates an affinity with target proteins, and finally forms a covalent bonds with sulfhydryl groups to exert cellular regulations and bioactivities. Previous studies have proved that a variety of electrophilic compounds can alleviate colitis in terms of disease activity index (DAI) scores, histopathological analysis and MPO activity, including curcumin [23], xanthohumol [24], cinnamaldehyde (CA) [25], SFN [26], ISO [27], and caffeic acid phenethyl ester [28]. Among these, SFN alleviated UC by regulating gut microbiota composition, increasing the content of volatile fatty acids and the expression of tight function proteins, and reducing pro-inflammatory cytokines, ISO attenuated DSS-induced colitis through the activation of nuclear transcription factor kappa B and the inhibition of the mitogen-activated protein kinase pathway and the caffeic acid phenethyl ester derivative-activated Nrf2/HO-1 pathway to alleviate DSS-induced colitis against oxidative stress. Although these FECs have thus far been reported to have activity in interfering with colitis, there is limited information for FECs with different structures on the relationship between the chemical structures and the effects in colitis. In this study, six representative FECs, which are relatively abundant and prevalent in daily foods, such as CA in cinnamon, SFN in broccoli, CGA in hawthorn, CA in wheat, and ISO in licorice, were selected to evaluate their effects in alleviating colitis in DSS-induced mice. The relationship between molecular characteristics and intervention effects was also explored.

## 2. Materials and Methods

### 2.1. Chemicals and Primary Detection Kits

FMA, CA, SFN, FA, and CGA were purchased from Shanghai Aladdin Bio-Chem Technology Co., Ltd. (Shanghai, China). ISO and DSS (molecular weight: 40,000) were purchased from Shanghai Macklin Biochemical Technology Co., Ltd. (Shanghai, China). Phosphate-buffered saline (PBS) was supplied by Hyclone (Logan, UT, USA). Total antioxidant capacity (T-AOC), superoxide dismutase (SOD), glutathione peroxidase (GSH-px), MPO, and malondialdehyde (MDA) were provided by Nanjing Jiancheng Bioengineering Institute (Nanjing, China). A tumor necrosis factor (TNF-α), interleukin (IL)-6, IL-1β ELISA kit was purchased from Xiamen Huijia Biotechnology (Xiamen, China). A BCA protein assay reagent kit was provided by Yeason Biotech Co., Ltd. (Shanghai, China). MonScript™ RTIII An All in One Mix reverse transcription kit and a MonAmp™ Fast SYBR^®^ Green qPCR Mix (NoneROX) PCR kit were purchased from Monad (Suzhou, China). Anti-Nrf2, anti-HO-1, anti-NQO1, and anti-β-actin were obtained from Cell Signaling Technology (Beverly, MA, USA). Goat anti-rabbit was purchased from Abcam (Cambridge, UK). The chemical structures of the FECs are shown in Figure 1.

### 2.2. Animals and Grouping

The experimental design was approved by the Institutional Animal Care and Use Committee of Jiangnan University (JN. No20210630c0640808 [203]). Sixty-four male C57BL/6 (18–20 g) mice were purchased from Jiangnan University Laboratory Animal Center. Mice were housed in a specific-pathogen-free (SPF) animal laboratory room kept at 25 ± 1 °C and exposed to a 12:12 light-dark cycle. Mice were acclimatized to the environment under standard conditions for a week with free access to tap water and a standard rodent diet. After acclimatization, mice were randomly divided into eight groups, in which the control (CON) and DSS groups were daily gavaged with 0.2 mL of PBS buffer, while the other six groups were daily gavaged with 20 mg/kg body weight of FMA, ISO, CA, FA, SFN, and CGA (dissolved in 0.2 mL PBS buffer) throughout the experiment, respectively. The drinking water of each group was replaced with 3% DSS solution during the colitis simulation period (7 d), except for the CON group. The experimental detail is shown in Figure 2. The DSS solution was changed every 2 d. At the end of the experiment, the mice were euthanized by cervical dislocation, the length of the colon was measured in each group, the colon tissue was removed as soon as possible, and the spleen was weighed. The colon tissues were quickly rinsed with ice-cold saline and frozen in liquid nitrogen for further analysis.

### 2.3. Evaluation of the DAI

The assessment of UC severity is based on the DAI score, according to previous research [29]. The DAI score of each mouse was recorded to assess the status of colitis disease, which was evaluated by a combination of indexes comprising the rate of weight change, degree of fecal formation, and blood in the stool. Scores were defined as follows: (i) percentage weight loss: 0 (0%), 1 (1–5%), 2 (5–10%), 3 (11–15%), 4 (>16%); (ii) stool formability: 0 (formed ball, granular), 2 (semi-formed, loose), 4 (unformed, watery); (iii) blood in the stool: 0 (no blood), 1 (weakly positive), 2 (positive), 3 (strongly positive), 4 (blood in the fresh stool).

### 2.4. Histopathological Analysis

A 1 cm long sample of the distal colon was taken and fixed in 4% paraformaldehyde. Paraffin sections were stained with hematoxylin and eosin (H&E). Colonic rings were visualized using a biological microscope (×200 magnifications).

### 2.5. Measurement of Inflammatory Cytokines and Antioxidant Levels

Inflammatory cytokines including TNF-α, IL-6, and IL-1β in the colon tissue were determined using commercial ELISA kits following the manufacturer’s instructions. The levels of T-AOC, GSH-px, SOD, MPO, and MDA in the colon tissue were measured using a commercial assay kit according to the manufacturer’s protocols.

### 2.6. RNA Isolation and Quantitative Real-Time PCR Analysis

Total RNA from the colon samples were determined using Trizol reagent, chloroform, isopropanol, and 75% ethanol solution. The concentration of each RNA sample was quantified using Nanodrop 2000 (Thermo Scientific, Waltham, MA, USA). The cDNA was synthesized with a MonScript™ RTIII All-in-One Mix according to the manufacturer’s instructions. Real-time PCR was performed using a MonAmp™ Fast SYBR^®^ Green qPCR Mix (NoneROX) PCR kit on a Monad Selected q225 Real-time PCR system (Monad, Suzhou, China). *β-actin* expression was used as an endogenous control. The PCR primers used are listed in Table 1.

### 2.7. Western Blotting

A sample of 50 mg colon tissue was homogenized with a protein lysate buffer (including proteinase inhibitor and phosphatase inhibitor) to extract the total protein, which was quantified by a BCA protein kit and diluted to the same protein concentrations. Equal amounts of the protein samples were separated by 10% SDS-PAGE and then transferred to PVDF membranes. The membranes were further blocked with 5% non-fat milk, and then incubated with anti-Nrf2, anti-HO-1, anti-NQO1, or anti-β-actin at 4 °C overnight. Next, they were incubated with a secondary antibody for 1 h [30]. After washing with TBST 3 times, images were visualized via the Imaging System (Tanon, Shanghai, China) and analyzed by ImageJ (NIH, Bethesda, MD, USA).

### 2.8. Calculation of Electrophilic Index, n-Octanol/Water Partition Coefficient (logP)

The density generalized function theory (DFT) method in the program Gaussian 09 (Gaussian Inc, Wallingford, CT, USA) was used to optimize the geometric structure of the investigated FECs at the B3LYP/6-31G* (d, p) level. The orbital energies of the highest occupied molecular orbital (HOMO) and orbital energies of the lowest occupied molecular orbital (LOMO) values of the corresponding FECs were subsequently calculated in the Gaussian program, and the electrophilic index (ω) was calculated using the following equations:ω = µ^2^/2η(1)
µ = (E_HOMO_ + E_LOMO_)/2(2)
η = E_LOMO_ − E_HOMO_(3)

The 2D structures of each FEC were downloaded from PubChem and then imported into the online database www.sioc-ccbg.ac.cn (2 March 2021) for log*P* value calculation [31].

### 2.9. Molecular Docking

The crystal structure of the Keap1 BTB domain (PDB ID: 4cxi) was obtained from the Protein Data Bank (PDB). The 2D structures of six FECs were obtained from PubChem and then converted into 3D structures by energy minimization in MOE. The binding pose of each compound to the active cysteine residues of Keap1 protein was docked using the MOE dock, and the binding affinity of each compound to the Keap1 BTB domain was predicted [32]. The drug binding pocket was used as a reference to select suitable and potential active sites. The free energy of binding of each compound to the Keap1 BTB domain was estimated using the GBVI/WSA score to find the optimal binding pose and the binding mode was analyzed under refinement minimization conditions.

### 2.10. Correlation Analysis

Pearson’s correlation analysis was performed between DAI scores of colitis mice and log*P* values, Keap1 affinity energy, and electrophilic index (ω) of the six FECs.

### 2.11. Statistical Analysis

All data are expressed as mean ± standard deviation (SD). The software GraphPad Prism 8.1 (GraphPad Software, La Jolla, CA, USA) and SPSS 20.0 (IBM Corporation, Armonk, NY, USA) was used to process and analyze the data using one-way ANOVA and Duncan’s post hoc test. For all data, *p* < 0.05 was considered statistically significant.

## 3. Results

### 3.1. FECs Supplementation Attenuated Colitis Differentially in DSS-Induced Mice

The mice in the DSS group showed a significant decrease in body weight compared to the CON group (*p* < 0.05), while the mice in each group supplemented with FECs improved their body weight loss (Figure 3A). As shown in Figure 3B, the DSS treatment led to a decrease of DAI score, but the supplementation of FECs resulted in a pronounced reduction of DAI scores in each group except the FMA and CGA groups. In particular, SFN supplementation significantly reduced the DAI score in DSS-induced colitis mice (*p* < 0.01). As shown in Figure 3C, the colons in the DSS group exhibited obvious congestion and severe curling. The colon length was shortened after DSS treatment which led to a 10.65% reduction compared with the CON group (Figure 3D). In contrast, supplementation with FECs inhibited the shortening of colon length. ISO, CA, FA, and SFN were much more effective than FMA and CGA in inhibiting the colon length shortening. As shown in Figure 3E, all FECs except CGA significantly ameliorated the increase of the spleen index due to the DSS induction. Taken together, ISO, CA, FA, and SFN supplementation effectively and notably mitigated colitis in DSS-induced mice, while FMA and CGA exhibited weak effects on colitis.

The sections of colon tissue were subjected to H&E staining to further reveal the effects of FECs supplementation on colonic histopathology (Figure 4). The colon tissue in the CON group was structured with tightly and orderly arranged colonic epithelial cells. On the contrary, the colon tissue in the DSS group exhibited the destruction of cells and glands in the colonic mucosal tissue (Figure 4B), including serious upper mucosa deformation (yellow triangle), crypt structure damage (red arrow), and inflammatory cells infiltration (red circle). As shown in Figure 4C–H, FECs provided different degrees of protection for colon cells and tissues in mice with colitis, but did not completely eliminate the damage induced by DSS. FECs supplementation greatly improved the epithelial barrier of the colon with well-structured crypts and villi, typically in the ISO and SFN groups of mice. Moreover, there was slighter submucosa edema (red star) and less inflammatory cells infiltration in FA-supplemented mice than those that were supplemented with FMA, CA, and CGA. According to the DAI score and histopathological analyses of colon, the investigated FECs exhibited intervention effects with a sequence of SFN, ISO > FA, CA > FMA, CGA.

### 3.2. Molecular Characteristic Calculation of FECs

The log*P* value, Keap1 affinity, and electrophilic index (ω) of investigated FECs are shown in Table 2. Log*P* value is a parameter that represents the transmembrane absorption capacity of a substance during the cellular absorption phase. The log*P* value of CGA and FMA were −0.42 and −0.34, respectively, whereas other FECs possessed log*P* values more than 0.

Furthermore, the Keap1 affinity energy of each FECs was calculated (Table 2). The smaller the Keap1 affinity energy, the stronger affinity of the electrophilic compound to the protein, which showed the highest affinity energy for SFN (−2.7 kcal/mol) and the lowest binding energy for CGA (−4.5 kcal/mol). As shown in Figure 5, the molecular docking conformation of each FECs was further analyzed. In the selected binding pockets, the investigated FECs primarily interacted with Tyr85, His129, Lys131, Val132, Cys151, and His154 of the pocket residues, which mainly formed hydrogen bond, van der Waals, Pi-cations, and Pi–Pi stacked forces (Appendix A). The hydroxyl, carbonyl, and sulfinyl groups in the FECs molecules would facilitate hydrogen bonds formation with amino acid residues, while the benzene ring could form Pi–Pi stacked, Pi-cation, and Pi-alkyl interactions. ISO and CGA showed similar affinity energy at −4.4 and −4.5 kcal/mol, perhaps due to their similar structures and interactions with pocket residues, including hydrogen bond, van der Waals, Pi–Pi stacked, Pi-cation, and Pi-alkyl, and also three pairs of hydrogen bonds with bond lengths in the range of 2.0–2.5 Å. FMA that was short of a benzene ring only showed hydrogen bond and van der Waals interactions with pocket residues, possessing affinity energy at −3.5 kcal/mol.

The electrophilic index (ω) measures the energy reduction due to the maximum electron flow between the ligand and the acceptor [33]. In general, the FECs with α,β-unsaturated carbonyl groups possessed higher electrophilic index (ω) than that of those that contained an isothiocyanate group, represented by ISO and SFN with electrophilic indexes (ω) of 2.35 and 1.11, respectively (Table 2). Among the FECs with α,β-unsaturated carbonyl groups, it seemed that the ketone and aldehyde had larger electrophilic indexes (ω) than the acid and esters.

### 3.3. FECs Altered the Levels of Inflammatory and Oxidative Indicators in DSS-Induced Colitis Mice

To further validate the differential effects of six FECs in colitis, we measured the levels of pro-inflammatory cytokines and oxidative indicators in colon tissues (Figure 6). After DSS induction, TNF-α, IL-1β, IL-6, MDA, and MPO in colon were significantly elevated, with *p* < 0.05, *p* < 0.01, respectively. Meanwhile, the DSS induction resulted in the significant decrease of T-AOC, GSH-px, and SOD activity. The six FECs showed various regulative effects on inflammatory and oxidative indicators, among which SFN exhibited significant regulation on all analyzed biochemical indexes, whereas FMA and CGA exhibited regulative effects without significance (Figure 6). ISO, CA, and FA supplementation exhibited significant regulation on T-AOC activity (*p* < 0.05), differential degrees of regulation on cytokines TNF-α, IL-1β, and IL-6, and slightly regulative effects on levels of MPO and MDA without significance. In particular, ISO significantly attenuated the levels of TNF-α and IL-1β (*p* < 0.05) and improved the level of SOD activity (*p* < 0.05). Both CA and FA significantly decreased the levels of IL-6 (*p* < 0.01), but slightly reduced the levels of TNF-α and IL-1β (*p* > 0.05). Although the six FECs regulated different aspects of inflammatory and oxidative factors to varying degrees, their regulation effectiveness was generally consistent with the results of the DAI score, histopathological analysis, and the log*P* values of the compounds.

### 3.4. FECs Supplementation Regulated the Nrf2 Pathway in the Colon of Mice with Colitis

Oxidative stress disrupts the physiological redox homeostasis, which leads to the production of pro-inflammatory factors in neutrophils and promotes the production of a large number of oxidation intermediates [34]. Nrf2 is a key transcription factor that regulates the expression of cytoprotective genes in response to oxidative stress. In this study, we further determined the Nrf2 signaling pathway-related mRNA and protein expression in colon tissue after FECs supplementation. As shown in Figure 7, compared with the CON group, the mRNA and protein expression levels of Nrf2, NQO1, and HO-1 were significantly reduced in the colon tissue of DSS-induced colitis mice. SFN and ISO supplementation significantly increased the expression levels of Nrf2 in colon tissues (*p* < 0.001), even more so than those of the CON group. Moreover, the downstream targets, HO-1 and NQO1, of Nrf2 signaling were also significantly enhanced. Although the other FECs showed weak improvement in mRNA expression levels of Nrf2 and the downstream targets, CA and FA significantly increased the protein expression levels of Nrf2 (*p* < 0.001) and elevated NQO1 and HO-1 expression, while FMA and CGA did not show adequate stimulation in the Nrf2 pathway. These results suggested that the investigated FECs activated the Nrf2 signaling pathway in the sequence of ISO, SFN > CA, FA > FMA, CGA as well.

### 3.5. Correlation of Molecular Characteristics of FECs with DAI Scores, Inflammatory and Oxidative Indicators, and Nrf2 Pathway in DSS-Induced Colitis Mice

Pearson’s correlation analysis was performed to reveal the relationship between the molecular characteristic values of FECs and their intervention effects in colitis (Appendix A). Among the six FECs, there was a strong negative correlation between log*P* values and DAI scores in the corresponding groups (r = −0.738, *p* = 0.047), indicating that the larger the log*P* value of the electrophilic compound, the lower the DAI score and the better improvement in colitis. Indeed, the DAI scores of colitis mice supplemented with FMA and CGA had log*P* values < 0 were approximate to those of mice in the DSS group. Moreover, there was a weak positive correlation between DAI scores and electrophilic index (r = 0.391, *p* = 0.222) and a weak negative correlation between DAI scores and Keap1 affinity energy (r = −0.592, *p* = 0.108). In addition, there was also a strong negative correlation between the electrophilic index of the FECs and Keap1 affinity energy (r = −0.754, *p* = 0.042).

Pearson’s correlation analysis between the colonic inflammatory, oxidative indicators, and the molecular characteristics of FECs was also conducted (Appendix A). Inflammatory indicators (TNF-α, IL-1β, IL-6) in the colon tissue showed a negative correlation with log*P* values (r = −0.544; r = −0.618; r = −0.577), but this was not significant. Moreover, a weak correlation between electrophilic index, Keap1 affinity energy and the molecular characteristic values of FECs was also observed. In the oxidative indicators, log*P* values showed a positive correlation with T-AOC, GSH-px, and SOD (r = 0.607; r = 0.621; r = 0.517), but showed a negative correlation with MPO and MDA (r = −0.371; r = −0.402), both of which were not significant. On the contrary, there was a strong positive correlation between MPO activity and electrophilic index (r = 0.747, *p* = 0.029).

Pearson’s correlation analysis was also performed between the protein level (Nrf2, NQO1, and HO-1) and the molecular characteristics of FECs. Appendix A show that there was a significant positive correlation between log*P* values and the protein level of Nrf2 (r = 0.922, *p* = 0.004). Log*P* values also showed a strong positive correlation with the protein level of NQO1 (r = 0.737, *p* = 0.047) and HO-1 (r = 0.759, *p* = 0.04), indicating that the larger the log*P* value of the electrophilic compound, the higher protein level of Nrf2 and downstream targets in colitis, and this result was consistent with the results of correlation analysis between log*P* values and DAI scores. However, there was a weak positive correlation between the protein level of Nrf2, NQO1, HO-1, and Keap1 affinity energy (r = 0.173; r = 0.468; r = 0.266) and a weak negative correlation between the protein level of Nrf2, NQO1, HO-1, and electrophilic index (r = −0.069; r = −0.382; r = −0.18).

## 4. Discussion

The present study aimed to reveal the differential effects of six prevalent FECs on colitis in DSS-induced mice and the underlying relationship with molecular characteristics. To our knowledge, this is the first study to report a comparison of FECs with different molecular profiles in rodents for their capability to alleviate colitis. Among the three representative molecular characteristics of the “penetration-affinity–covalent binding” procedure, log*P* values were found to be highly correlated with DAI scores in the FECs supplemented groups. Moreover, the mechanism of FECs in alleviating colitis was further confirmed to be through activating Nrf2 signaling and improving antioxidant defense system to ameliorate oxidative stress and inflammation.

The molecular descriptors of a compound, i.e., molecular characteristic values, can be classified into many types depending on the different data types of the post-computational system. However, not all molecular descriptors are useful and may contain duplicate information. Therefore, the extraction of molecular feature values is especially important; it is an indispensable step in image analysis and processing, which can effectively eliminate redundant information and reduce the number of features entered in the database [35]. Research has showed that, in addition to the electrophilic properties of electrophilic compounds, the molecular characteristics of the substance itself, such as the lipid–water partition coefficient, spatial site resistance, and solubility of electrophilic compounds, also affect the rate of forming covalent adducts with the target [36]. In addition, molecular theoretical calculations revealed that the calculated log*P* and electrophilic index were the main characteristics to determine the anti-human immunodeficiency virus type 1 integrase activity of flavonoids [37]. Therefore, in this study, molecular characteristics such as the log*P* value, electrophilic index, and the ability of the electrophilic compound bound to the target protein were the focus of attention in the calculation. The evaluation of the affinity binding of FECs to Keap1 protein was performed by molecular docking. Molecular docking is a technical method that is widely used to simulate the interaction of ligands with receptor proteins. It can offer the following two kinds of important information: (i) potential binding sites of ligands and receptors; (ii) the binding affinity and binding pose, to analyze which types of interactions are crucial in forming protein–ligand complex, and docking scores to assess the binding potential of the protein–ligand complex [38]. The strength of the interaction between FECs and the target protein was judged by the affinity energy. As per the results of molecular docking, when the affinity energy between the ligand and the receptor was stronger, such as ISO and CGA, they were, to an extent, more likely to grab the active pocket of Keap1. As shown in ISO, if the FECs in the active pocket had high electrophilic index, they were prone to form covalent bonds with cysteine residues on the basis of non-covalent interactions and show strong cellular signaling regulation.

Many reports are available on the utility of the natural compounds studied in this report on colitis. The cytoprotective properties of FECs may be related to the activation of the Nrf2 pathway. Yao et al. showed by Western blotting and RT-PCR that xanthohumol in the presence of electrophilic groups activated Nrf2 expression and upregulated the downstream target genes NQO1 and HO-1 to enhance antioxidant capacity and reduce the propensity of tissues to develop into other diseases [39]. Dimethyl fumarate (DMF), which has a similar structure to FMA, was reported by Liu et al. to decrease MPO activity and induce the Nrf2/ARE pathway in colitis by upregulating Nrf2 protein levels, as well as mRNA and protein levels of Nrf2 target genes NQO1 and HO-1 and promoting Nrf2 nuclear translocation, which plays a protective role in colitis [40]. MPO is a peroxidase specifically produced by neutrophils, participating in the enzymatic oxidation and elimination of invading microorganisms in phagocytes. The levels of MPO activity in intestinal epithelial tissue reflected its oxidative stress and neutrophils infiltration. In this study, each FECs supplementation obviously decreased the levels of MPO activity in colon. The correlation analysis between MPO activity and the molecular characteristics of FECs showed that electrophilic index was the main factor affecting the MPO activity. However, the underlying relationship between them still requires further research. Recent studies have shown that the protective effect of SFN against colitis is dependent on the expression of Nrf2 as well as phase II enzymes [26]. However, not all FECs supplementation groups in this study had a significant protective effect against colitis. As shown in Figure 3B, not all FECs showed improvement in the DSS-induced DAI scores in colitis. The DAI score of the CGA group was higher than that of the DSS group, and the symptoms of colitis were not significantly improved. Combined with the log*P* value of CGA and findings of previous studies, this may be related to its high water partition rate and inadequate supplementation dose. However, most of the FECs activated the Keap1-Nrf2 pathway and mediated the transcriptional expression of a series of cytoprotective proteins (phase II enzymes) that exerted antioxidant and detoxifying effects. These proteins are catalytic, not easily consumed, have a relatively long half-life and frequently lead to a detoxification reaction [41]. In the correlation analysis, log*P* value was found to be the main influencing factor for FECS to activate Nrf2 and mediated the expression of downstream proteins (NQO1, HO-1), which explains why FECs with low log*P* value have a poor effect in ameliorating colitis. This may be because low log*P* value makes FECs unable to penetrate the colon or cell and interact with the Keap1 protein in cells, thus affecting the activation of the Nrf2 pathway. At the same time, this also reflects that Nrf2 and its downstream series of cytoprotective proteins could play a role in alleviating colitis. Our results showed that significant inhibition of Nrf2 and its downstream genes NQO1 and HO-1 was observed in DSS-induced colitis mice. In contrast, supplementation with the same dose of FECs, especially ISO and SFN, significantly improved DSS-induced colitis in mice by increasing the mRNA and protein expression levels of Nrf2, NQO1, and HO-1. Wagner et al. also found that SFN pretreatment increased the expression of Nrf2 related genes in colitis mice [42], which was consistent with our results in this study. In the present study, this point contributes to the understanding of the biological activity of FECs alleviating colitis in relation to the activation of the Keap1–Nrf2 pathway.

This study suggested that FECs with high log*P* value have stronger bioactivity in ameliorating colitis than FECs with low log*P* value. We chose log*P* value as an indicator of the cellular permeability of the compound. A previous study suggested that there was a link between log*P* value of compounds and their cytotoxicity [43]. The cytotoxicity of 11 thiourea derivatives was found to be positively correlated with log*P* value for cancer cells and negatively correlated with log*P* value for HACAT normal cells [44]. In the present study, no toxic effects of FECs were observed in mice, which may be related to the low dose of the supplementation. In this study, a correlation between the mitigative effect in colitis and the compound log*P* value was revealed. The larger the log*P* value, the higher the expression of Nrf2, NQO1, and HO-1, and the higher the mitigative effect in colitis. However, there is a maximum and most fitting point of log*P* value, which represents the highest biological activity, and once this point is exceeded, the corresponding biological activity produced will be reduced [45]. Studies on the screening of the antiviral activity of indigo derivatives indicate that the ideal range of log*P* value is between 1.0 and 3.0 [46]. Another study compared the different structures of tannins in relation to their antioxidant activity and found that log*P* value was the main property predicting the antioxidant capacity of the phenolic compounds analyzed [47]. The QSAR study of antioxidant activity of wine polyphenols also found that the antioxidant activity of flavonoids increased with the increase of log*P* value [48]. In cellular experiments, antioxidants with high log*P* value have stronger cytoprotective effects [49]. These previous findings are consistent with the results in the present study. The log*P* value of FECs showed a high correlation with their alleviating effects in colitis, including inflammatory factors, oxidative indicators, and Nrf2 signaling pathway in colon tissue and cells. The high log*P* value indicates that the compound is lipophilic, which would facilitate the permeation into cells, reach the pocket residues of Keap1, and then generate an affinity–electrophilic reaction process with sulfhydryl groups to activate Nrf2 pathway, thus exhibiting a strong cytoprotective effect in colitis. In addition, the finding of the correlation between the protein level (Nrf2, NQO1, and HO-1) and the molecular characteristics of FECs also verified that log*P* value affected the expression level of Nrf2 protein. In terms of the weak alleviating effect of CGA supplementation in DSS-induced mice, this may be related to its negative log*P* value, although it has high Keap1 affinity and electrophilic index, which would impede the absorption and utilization of CGA. A previous study has revealed that more than half of CGA intake was decomposed and transformed into hippuric acid and other metabolites by microorganisms in the gut [50], but with a poor absorption of CGA in the small intestine. Among the six investigated FECs, the highest Keap1 affinity energy and lowest electrophilic index of SFN notwithstanding, it exhibited the strongest improvement effect in colitis. This implies that Keap1 affinity energy and electrophilic index had less effect on the bioactivity of FECs to alleviate colitis as long as they were within an appropriate range, while the permeability represented by log*P* value was an important prerequisite for the bioactivity of FECs.

## 5. Conclusions

In conclusion, distinct alleviating effects in DSS-induced colitis mice were revealed in six prevalent FECs, in which ISO and SFN supplementation exhibited the strongest effectiveness. FECs regulated the oxidative stress and inflammatory response in colon by activating the Keap1–Nrf2 pathway, which served to alleviate colitis. The log*P* values, rather than the Keap1 affinity energy and electrophilic indexes of FECs, showed a strong correlation with colitis improvements. This suggested that the bioactivity of FECs was influenced not only by the inherent electronic properties, but also by the permeability represented by log*P* values. However, further research needs to examine the fitting range of log*P* value for FECs and whether excessive log*P* values of FECs would counteract their colitis improvement effects.

## Figures and Tables

**Figure 1 antioxidants-11-02406-f001:**
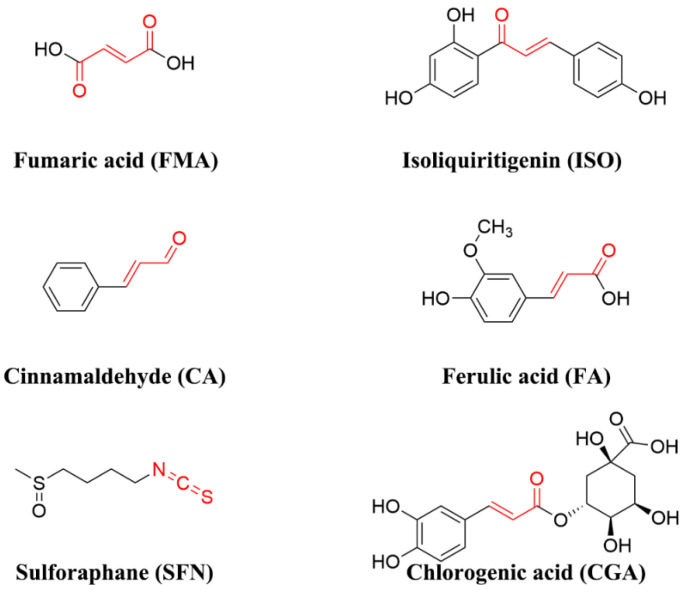
Structures of the investigated FECs.

**Figure 2 antioxidants-11-02406-f002:**
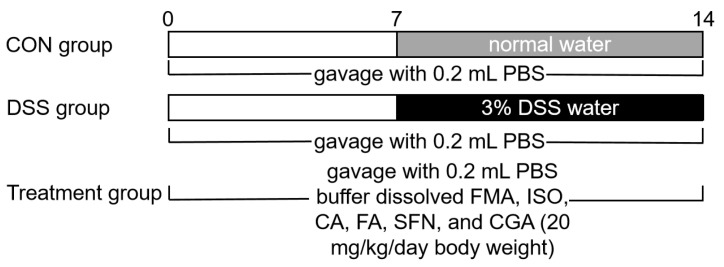
Flow chart of experimental design.

**Figure 3 antioxidants-11-02406-f003:**
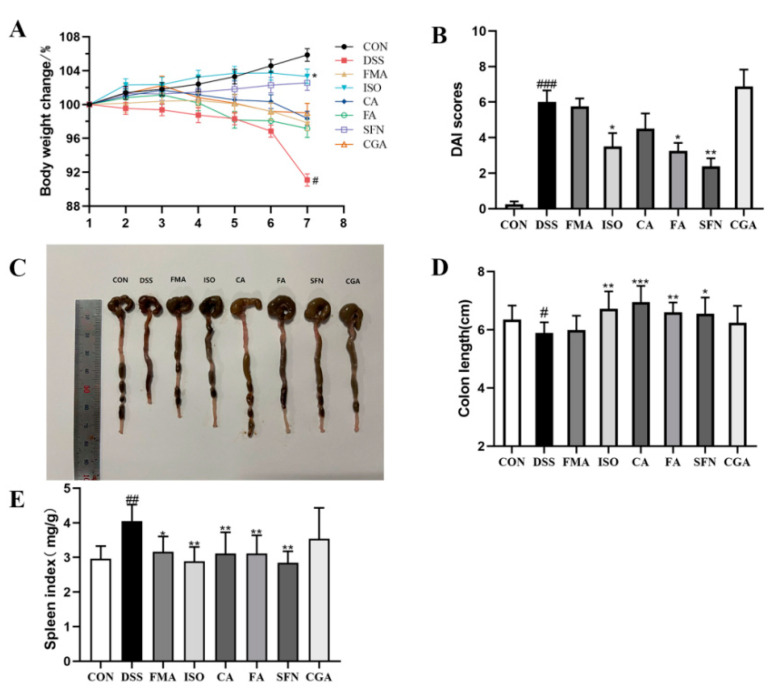
Effects of FECs on weight change, DAI score, colonic shortening and splenic index in DSS-induced mice. (**A**) Body weight changes; (**B**) DAI scores at the end; (**C**) Images of the colon length; (**D**) Colon length; (**E**) Spleen index. Data are expressed as mean ± SD (*n* = 8). ^#^ *p* < 0.05, ^##^
*p* < 0.01, ^###^
*p* < 0.001, compared with the CON group; * *p* < 0.05, ** *p* < 0.01; *** *p* < 0.001, compared with the DSS group.

**Figure 4 antioxidants-11-02406-f004:**
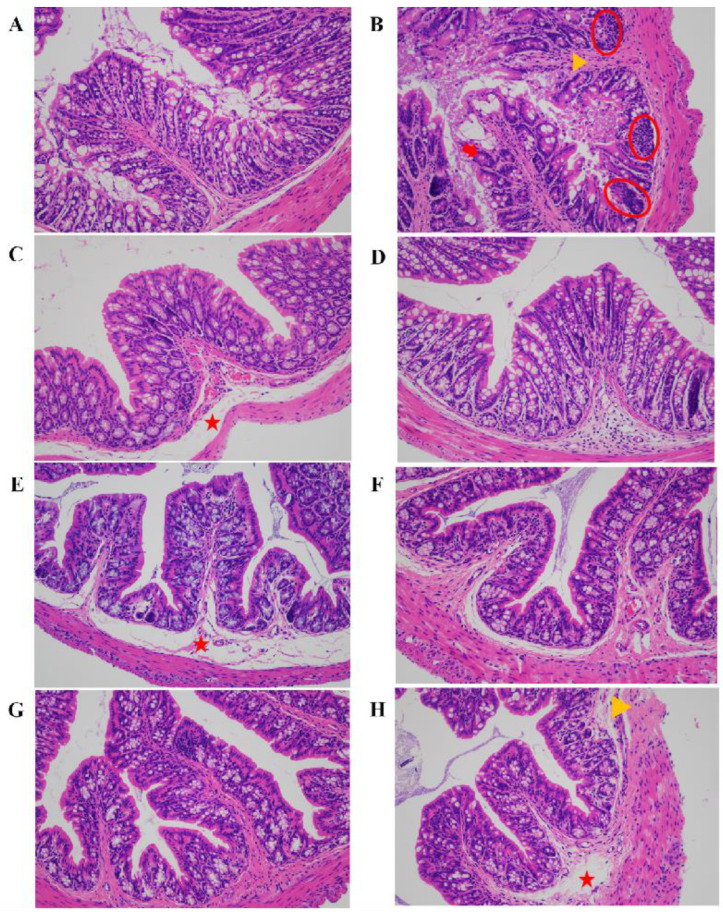
Effect of FECs supplementation on histopathological changes by H&E staining in DSS-induced mice colon, ×200 magnifications. (**A**) CON group; (**B**) DSS group; (**C**) FMA group; (**D**) ISO group; (**E**) CA group; (**F**) FA group; (**G**) SFN group; (**H**) CGA group. Inflammatory cells infiltration (red circle), crypt structure damage (red arrow), upper mucosa deformation (yellow triangle), submucosa edema (red star).

**Figure 5 antioxidants-11-02406-f005:**
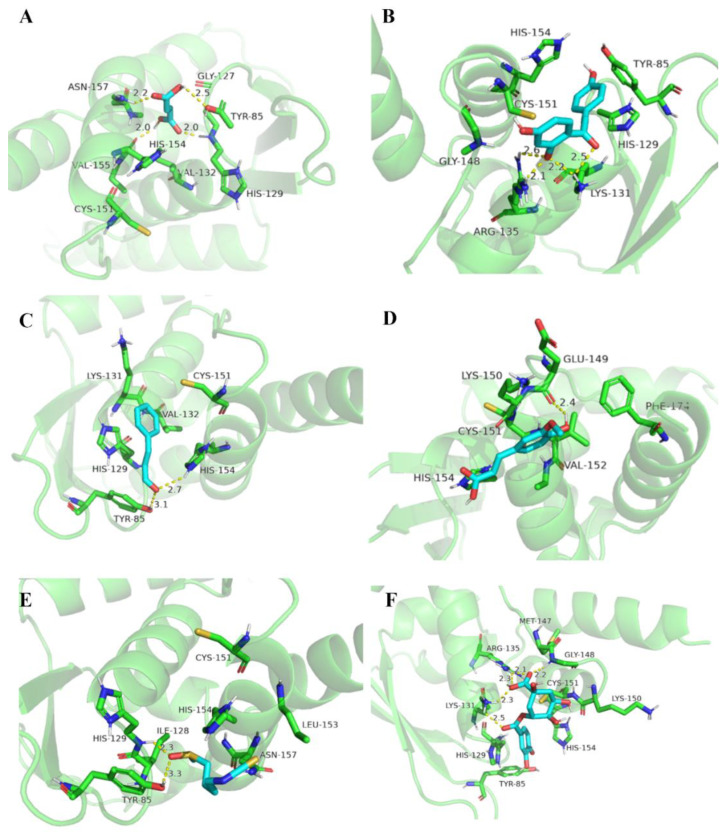
Molecular docking of FECs and BTB domain of Keap1. (**A**) FMA; (**B**) ISO; (**C**) CA; (**D**) FA; (**E**) SFN; (**F**) CGA.

**Figure 6 antioxidants-11-02406-f006:**
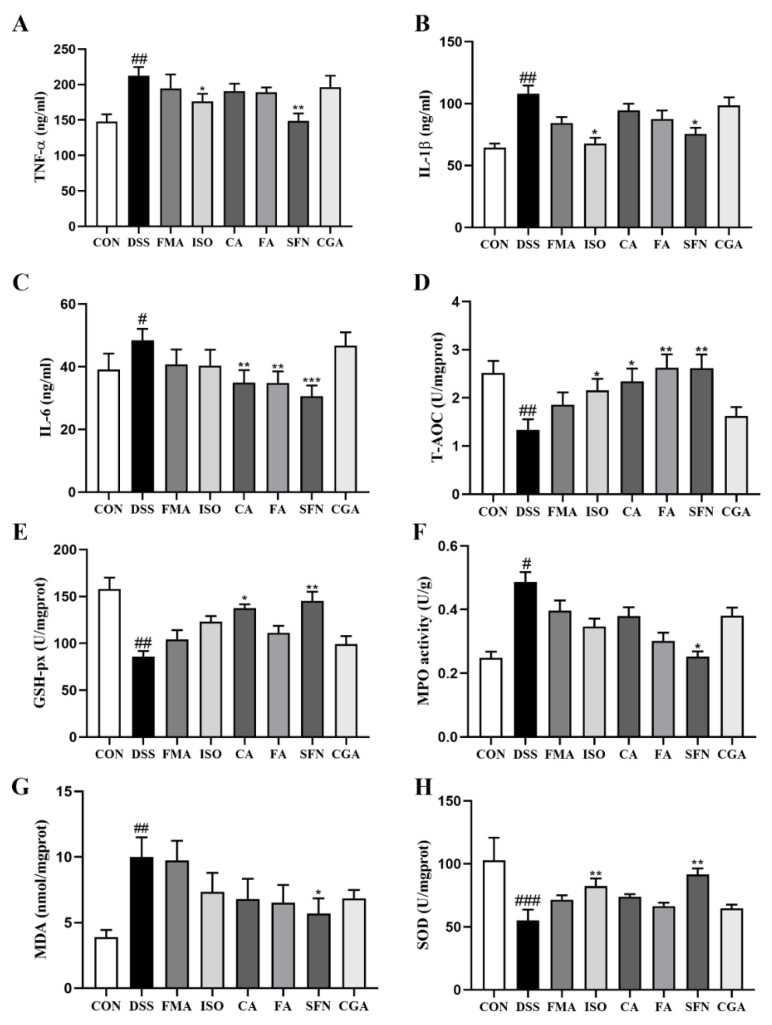
Effect of FECs on inflammatory and oxidative factors in colon tissue of mice with colitis. (**A**) TNF-α; (**B**) IL-1β; (**C**) IL-6; (**D**) T-AOC; (**E**) GSH-px; (**F**) MPO; (**G**) MDA; (**H**) SOD. Data are expressed as mean ± SD (*n* = 8). ^#^
*p* < 0.05, ^##^
*p* < 0.01, ^###^
*p* < 0.001 compared with the CON group; * *p* < 0.05, ** *p* < 0.01; *** *p* < 0.001, compared with the DSS group.

**Figure 7 antioxidants-11-02406-f007:**
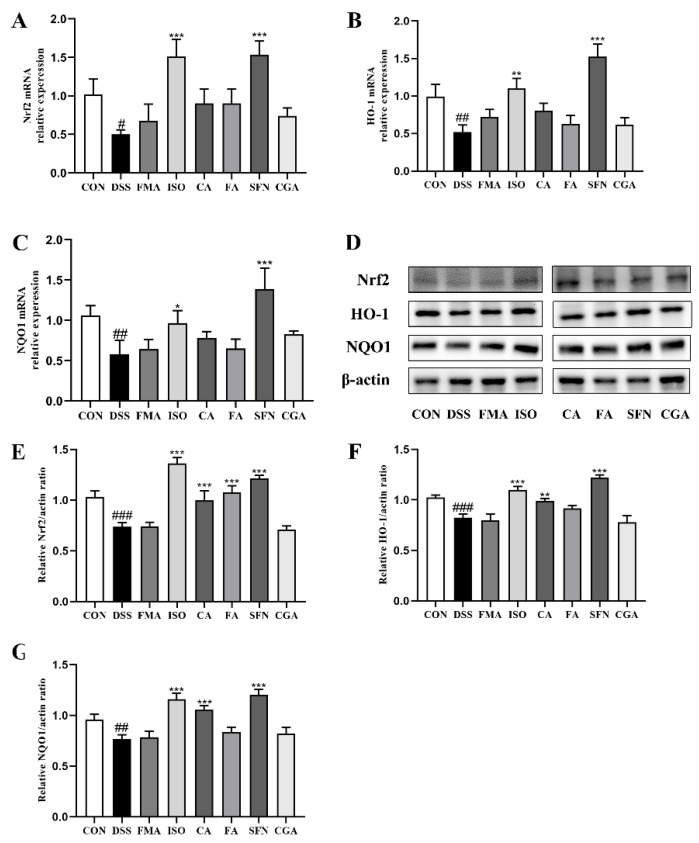
The mRNA and protein expression levels of the Nrf2 pathway in colon tissue of mice with colitis. mRNA expression of (**A**) Nrf2; (**B**) NQO1; (**C**) HO-1; (**D**) Western blotting of Nrf2, HO-1 and NQO1; Relative protein expression of (**E**) Nrf2; (**F**) HO-1; (**G**) NQO1. Data are expressed as mean ± SD (*n* = 8). ^#^
*p* < 0.05, ^##^
*p* < 0.01, ^###^
*p* < 0.001, compared with the CON group; * *p* < 0.05, ** *p* < 0.01; *** *p* < 0.001, compared with the DSS group.

**Table 1 antioxidants-11-02406-t001:** Sequence of the primers.

Gene	Forward Primer	Reverse Primer	Accession Number
β-actin	GGCTGTATTCCCCTCCATCG	CCAGTTGGTAACAATGCCAT	NM_007393
Nrf2	TCTTGGAGTAAGTCGAGAAGTG	GTTGAAACTGAGCGAAAAAGGC	NM_010902
NQO1	AGGATGGGAGGTACTCGAATC	AGGCGTCCTTCCTTATATGCT	NM_008706
HO-1	AAGCCGAGAATGCTGAGTTCA	GCCGTGTAGATATGGTACAAG	NM_010442

**Table 2 antioxidants-11-02406-t002:** Molecular characteristic values of FECs.

Compound Name	Log*P* Value	Keap1 Affinity Energy (kcal/mol)	Electrophilic Index (ω)
FMA	−0.34	−3.5	1.92
ISO	3.18	−4.4	2.35
CA	1.9	−3.6	2.31
FA	1.51	−3.5	1.87
SFN	1.41	−2.7	1.11
CGA	−0.42	−4.5	1.97

## Data Availability

Data are contained within the article and Appendix A.

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
