# Peer review of "The Alleviation of Dextran Sulfate Sodium (DSS)-Induced Colitis Correlate with the logP Values of Food-Derived Electrophilic Compounds"

_antioxidants, 2022, doi:10.3390/antiox11122406_

Round 1

Reviewer 1 Report

Interesting study showing positive influence of food-derived electrophilic compounds on dextran sulfate 2 sodium (DSS)-induced colitis in mice.

Authors performed analysis of six FEC's, from which sulforaphane (SFN) and isoliquiritigenin (ISO) showed high logP values in successfully improved DSS-induced colitis outcome.

Authors used 20 mg/kg body weight of FMA, ISO, CA, FA, SFN, and 131 CGA dissolved in 0.2 mL PBS buffer. Why this concentration was chosen? Another concentration has been tested? What about cytotoxicity of used supplementation scheme?

Is seven days period enough for obtaining statistical significant data?

Why Authors choose this six food-derived electrophilic compounds?

Reviewer 2 Report

The article presented by Xiang-Rong Cheng and collaborates, entitled “Food-derived electrophilic compounds alleviate dextran sulfate sodium (DSS)-induced colitis correlated with logP values”, is an original article to assess to investigate the differential effects of six prevalent Food-derived electrophilic compounds (FECs; Fumaric acid (FMA), isoliquiritigenin (ISO), cinnamaldehyde (CA), ferulic acid (FA), sulforaphane (SFN), and chlorogenic acid (CGA)) on colitis in dextran sodium sulfate (DSS)-induced mice and the underlying relationship with molecular characteristics. The methods used to make the assertions need to be worked out in depth

Major revision:

1.       The final objective of the introduction is to carry out a small review where the state of the art of the research to be carried out is explained. The article has serious shortcomings:

a.       Line 98. Although some FECs have thus far been reported to have activity in interfering with colitis. The authors must expose the knowledge up to the moment of the matter. They need to put all the references that talk about the FECS in CU.

b.       Line 100. six prevalent FECs were selected…. The authors must justify the choice taking into account what has been described in the literature.

2.       Line 124. 64 male C57BL/6 (18-20 g). In the 21st century, it is difficult to understand why experiments are still being done only with males. The authors must justify and take it into account for future work since there is no negative influence of hormonal cycles (denied by various studies REBECCA M. SHANSKY, 2019)

3.       Line 233. FECs supplementation greatly improved the epithelial barrier of colon with well-structured crypts and  villi, typically in ISO and SFN groups of mice… The authors must mark the images with arrows, stars or dots the findings that they indicate in the text. In general, photo B, which corresponds to DSS, does not show the mucosa with great damage (for the one described in figure 2b). The rest of the photos do not represent damage either and no correlation is observed with figure 2b, they seem to be almost like the control rather than the DSS group.

4.       Line 291. Since the DAI data are subjective (and in view of the IH chosen that do not correlate with Figure 2b), it is necessary to perform the correlations of molecular characteristics of FECs with the data obtained from the ELISA and mRNA to support the conclusion.

Minor revision:

1.       Line 46. ARE-related genes. Explain

2.       Line 123. The authors should detail the protocol, both for colitis and for treatment with FECS. Only treated for 7 days with DSS? It is not clear when they start treatment with FECS. An explanatory drawing would improve understanding. Is there no vehicle group? Leaves?

3.       Line 139. Evaluation of the disease activity index (DAI). Was performed blindly?. They must specify.

4.       Line 152. Measurement of inflammatory cytokines and antioxidant levels. Where were the measurements made? blood? secretomas?

5.       Line 242. Effect of FECs supplementation on histopathological changes by H&E staining in DSS induced mice colon. It improves the reading if it is introduced in the small images read from the FECS in question

6.       Line 278. Correlation of molecular characteristics of FECs with DAI scores in DSS-induced colitis mice. The data from correlations should be displayed in a table

Round 2

Reviewer 2 Report

The authors have introduced the suggested changes